# High-throughput discovery of genetic determinants of circadian misalignment

**Tao Zhang**[1⊙], **Pancheng Xie**[1⊙], **Yingying Dong**[1⊙], **Zhiwei Liu**[1], **Fei Zhou**[1], **Dejing Pan**[1], **Zhengyun Huang**[1], **Qiaocheng Zhai**[1], **Yue Gu**[1], **Qingyu Wu**[2,3], **Nobuhiko Tanaka**[4], **Yuichi Obata**[4], **Allan Bradley**[5], **Christopher J. Lelliott**[5], **Sanger Institute Mouse Genetics Project**[¶], **Lauryl M. J. Nutter**[6], **Colin McKerlie**[6], **Ann M. Flenniken**[6], **Marie-France Champy**[7], **Tania Sorg**[7], **Yann Herault**[7], **Martin Hrabe De Angelis**[8,9], **Valerie Gailus Durner**[8], **Ann-Marie Mallon**[10], **Steve D. M. Brown**[10], **Terry Meehan**[11], **Helen E. Parkinson**[11], **Damian Smedley**[12], **K. C. Kent Lloyd**[13], **Jun Yan**[14], **Xiang Gao**[14], **Je Kyung Seong**[15], **Chi-Kuang Leo Wang**[16], **Radislav Sedlacek**[9], **Yi Liu**[17], **Jan Rozman**[8,9,18]*, **Ling Yang**[1]*, **Ying Xu**[1,3]*

1 Cambridge-Suda Genomic Resource Center, Jiangsu Key Laboratory of Neuropsychiatric Diseases, Medical college of Soochow University, Suzhou, Jiangsu, China, 2 Cyrus Tang Hematology Center, Collaborative Innovation Center of Hematology, Soochow University, Suzhou, China, 3 State Key Laboratory of Radiation Medicine and Prevention, Medical college of Soochow University, Suzhou, China, 4 RIKEN BioResource Center, Tsukuba, Japan, 5 The Wellcome Trust Sanger Institute, Wellcome Genome Campus, Hinxton, United Kingdom, 6 The Centre for Phenogenomics, Toronto, Canada, 7 CELPHEDIA, PHENOMIN, Institut Clinique de la Souris (ICS), Illkirch, France, 8 German Mouse Clinic, Institute of Experimental Genetics, Helmholtz Zentrum München, German Research Center for Environmental Health, Munich, Germany, 9 Czech Centre for Phenogenomics, Institute of Molecular Genetics of the Czech Academy of Sciences, Vestec, Czech Republic, 10 Medical Research Council Harwell Institute (Mammalian Genetics Unit and Mary Lyon Centre), Harwell, United Kingdom, 11 European Molecular Biology Laboratory, European Bioinformatics Institute, Hinxton, United Kingdom, 12 School of Medicine and Dentistry, Queen Mary University of London, London, United Kingdom, 13 School of Medicine and Mouse Biology Program, University of California, Davis, California, United States of America, 14 SKL of Pharmaceutical Biotechnology and Model Animal Research Center, Collaborative Innovation Center for Genetics and Development, Nanjing Biomedical Research Institute, Nanjing University, Nanjing, China, 15 College of Veterinary Medicine, Seoul National University, and Korea Mouse Phenotyping Center, Seoul, Republic of Korea, 16 National Laboratory Animal Center, National Applied Research Laboratories (NARLabs), Taipei, Taiwan, 17 Department of Physiology, University of Texas Southwestern Medical Center, Dallas, Texas, United States of America, 18 German Center for Diabetes Research (DZD), Neuherberg, Germany

⊙ These authors contributed equally to this work.
¶ Membership of the Sanger Institute Mouse Genetics Project is listed in the Acknowledgments.
* jan.rozman@img.cas.cz (JR); lyang@suda.edu.cn (LY); yingxu@suda.edu.cn (YX)

**Data Availability Statement:** The data underlying the results presented in this study are available from the IMPC consortium (https://www.mousephenotype.org/help/api-access/) or Cambridge-suda genomic resource center (http://

## Abstract

Circadian systems provide a fitness advantage to organisms by allowing them to adapt to daily changes of environmental cues, such as light/dark cycles. The molecular mechanism underlying the circadian clock has been well characterized. However, how internal circadian clocks are entrained with regular daily light/dark cycles remains unclear. By collecting and analyzing indirect calorimetry (IC) data from more than 2000 wild-type mice available from the International Mouse Phenotyping Consortium (IMPC), we show that the onset time and peak phase of activity and food intake rhythms are reliable parameters for screening defects of circadian misalignment. We developed a machine learning algorithm to quantify these two parameters in our misalignment screen (SyncScreener) with existing datasets and used it to screen 750 mutant mouse lines from five IMPC phenotyping centres. Mutants of five

gofile.me/2F1pE/RP2URKxV2). Numerical data that underlying graphs or summary statistics are provided in spreadsheet form as Supporting Information.

**Funding:** The funders had no role in study design, data collection and analysis, decision to publish, or preparation of the manuscript" to the end of Funding information as: "This work was supported by grants from the Ministry of Science and Technology (2018YFA0801100 to YX) and the National Natural Science Foundation of China (31630091 to Y.X, 31871185 to Y.D., 31600958 to Z.L., 11671417 to L.Y.). We also thank the Priority Academic Program Development of the Jiangsu Higher Education Institutes (PAPD) and National Center for International Research (2017B01012). AMM, TFM, DS, HP, PF and the IMPC Data Coordination Centre are supported by the NIH Common fund grant (UM1HG006370). Infrafrontier grant 01KX1012, support by the German Center for Diabetes Research (DZD), EU Horizon2020: IPAD-MD funding 653961 (MHA). The funders had no role in study design, data collection and analysis, decision to publish, or preparation of the manuscript.

**Competing interests:** The authors have declared that no competing interests exist.

genes (*Slc7a11*, *Rhbdl1*, *Spop*, *Ctc1* and *Oxtr*) were found to be associated with altered patterns of activity or food intake. By further studying the *Slc7a11tm1a/tm1a* mice, we confirmed its advanced activity phase phenotype in response to a simulated jetlag and skeleton photoperiod stimuli. Disruption of *Slc7a11* affected the intercellular communication in the suprachiasmatic nucleus, suggesting a defect in synchronization of clock neurons. Our study has established a systematic phenotype analysis approach that can be used to uncover the mechanism of circadian entrainment in mice.

## Author summary

Synchronization to environmental changes such as day and night cycles and seasonal cycles is critical for survival. Organisms have therefore evolved a specialized circadian system to anticipate and adapt to daily changes in the environment. Loss of synchrony between the internal circadian clock and environment day and night changes is responsible for jet lag, but may also promote sleep disorders, metabolic disorders and many diseases. The availability of large amounts of mouse data from the International Mouse Phenotype Consortium provides new opportunities to identify novel genetic components of mouse behaviour and metabolism. In this study, we performed a high-throughput identification of genetic components of circadian misalignment by developing a machine learning-based algorithm. By analyzing the indirect calorimetry parameters from more than 2000 C57BL/6N mice and mice from 750 mutant lines, we identified 5 genes involved in circadian misalignment of activity and feeding behaviour. Further analyzing genetic knock-out mice for one of these genes, we were able to validate our screening method by functional studies. Our systemic analysis thus paves the way for searching the genetic determinants for circadian misalignment.

## Introduction

The circadian clock is one of the best-characterized mechanisms that can mediate the influence of environmental cues on molecular, physiological and behavioural activities in almost all organisms. The suprachiasmatic nucleus (SCN) is the central circadian pacemaker in mammals that receives photic information via the retina, integrates time-related information of tissues and organs, and then transmits timing information to cells and tissues to regulate physiology and behaviour to entrainment of animals to the daily changes of environmental cues [1,2]. Chronic misalignment between the circadian clock and the environment has been implicated in many pathological processes such as sleep disorders, cardiovascular diseases, metabolic disorders, and cancer [3,4]. Mice with a defective light input pathway (lacking rods, cones, and melanopsin (*Opn4-/- Gnat1-/- Cnga3-/-*)), cannot be entrained to light/dark cycles [5,6]. On the other hand, arrhythmicity at the tissue or behavioural level and attenuated light-induced phase shift can result from impaired expression of coupling peptides in the SCN, such as genetic ablation of vasoactive intestinal peptide (VIP), Gastrin-releasing peptide (GRP) or VIP receptors VPAC in mice [7–9]. In humans, dysfunction or misalignment of the circadian clock with environmental cues alters the timing of the sleep-wake cycle [10–12]. Mice with mutations orthologous to the human mutations (PER2S662G, CK1δT44A) recapitulate human phase-advanced behavioural rhythms and transgenic mice carrying PER1S714G mutation advances feeding behaviour [13–16], indicating that mice are a good model for human

circadian functions. In addition, activity, feeding, temperature and glucocorticoid signals can also affect the circadian phase [17–21]. These are all zeitgebers of the circadian clock and will impart phase information on their target tissues [22]. Circadian misalignment is a consequence of conflicting signals of these zeitgebers.

The International Mouse Phenotyping Consortium (IMPC) project is generating a genome-wide annotation of gene functions by systematically generating and phenotyping a collection of standardized single-gene knockout mice [23–25]. Indirect calorimetry (IC) datasets are collected by the IMPC pipelines with standardized protocols (https://www. mousephenotype.org/impress/protocol/86). Activity parameters are monitored using a metabolic chamber equipped with infrared beam instead of using a running wheel to avoid artificially enhanced or weakened activity. A food intake monitoring system is also integrated to investigate diurnal patterns of feeding rhythms and behaviours. The availability of the IC datasets from mice generated and phenotyped by the IMPC offered a unique opportunity to perform a large-scale screen for mutants with defective circadian misalignment. By analyzing the IC data from more than 2,000 wild-type mice available from the IMPC, we identified two reliable parameters of circadian misalignment. We developed a machine learning algorithm for circadian parameter recognition (SyncScreener). Using this algorithm, we screened mice from 750 mutant lines from five IMPC centres and identified five novel genes involved in circadian misalignment. Among these genes, the function of *Slc7a11* in circadian entrainment was confirmed by creating a knockout mouse, demonstrating that our approach is effective in uncovering mechanistic insights into circadian entrainment.

## Results

### Analyzing mouse indirect calorimetry datasets to identify reliable parameters for circadian misalignment

Misalignment between different components of the circadian systems and sleep-wake cycles or food intake behaviour as a result of genetic, environmental or behavioural factors might be an important contributor to diseases as well as how we can treat and prevent them. The mouse mutants with circadian misalignment would result in phase changes in locomotor activity and/or food intake behaviour under the light/dark cycle. We attempt to provide a systematic approach to identify genes underlying misalignment between circadian systems and behavioural factors. We thus collected available IC data from five IMPC centres: the Riken BioResource Center (RBRC), The Centre for Phenogenomics (TCP), the Institut Clinique de la Souris (ICS), the Wellcome Trust Sanger Institute (WTSI) and the Helmholtz Zentrum Munich (HMGU) and designed a workflow for systematic and unbiased analysis of circadian misalignment for more than 2,000 wild-type C57BL/6N mice and mice from 750 mutant mouse lines (Fig 1).

Baseline data from wild-type C57BL/6N were used to compare the reliability of data between centres and/or within centres to choose robust parameters of circadian misalignment. We first analyzed the activity/food intake cycle data for wild-type mice from the five IMPC centres. The total activity and food intake for each mouse was calculated and expressed as 1-hour averages over at least 21 hours according to the IMPC protocol. As expected, both activity and food intake rhythms displayed two clear peaks: a strong early evening (E) peak and a weak early morning (M) peak [26,27] (Fig 2A and 2B). The onset time at which mouse activity or food intake behaviour transitioned from the resting state to the active state could be clearly seen. These observations suggest that the onset time and E peak phase are potentially reliable parameters for circadian behaviour despite the diversity of data from different centres.

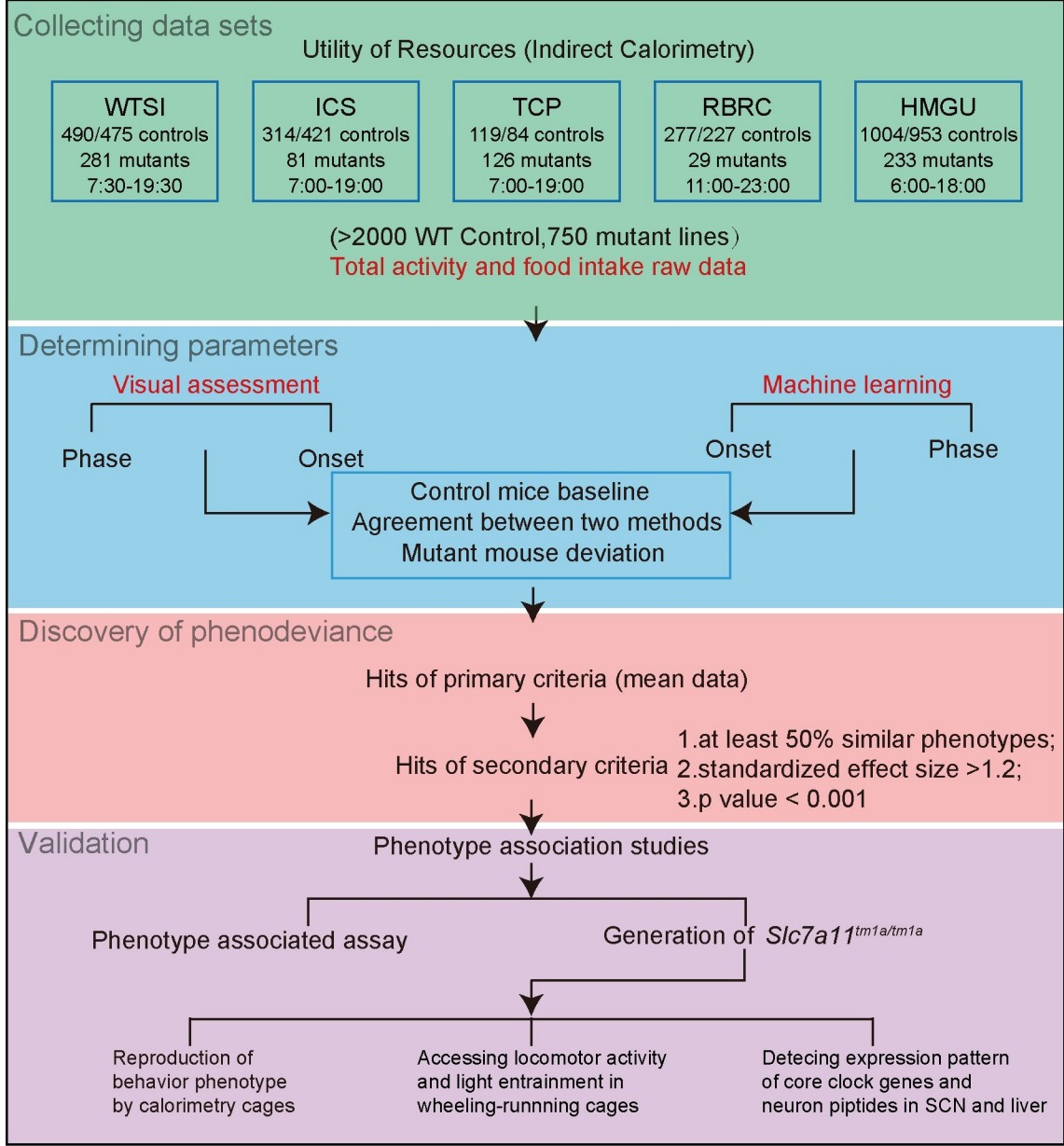

**Fig 1. Screening strategy diagram.** IC data sets were collected from five centres (WTSI, ICS, TCP, RBRC and HMGU). Baseline data from >2000 wild-type C57BL/6N mice were used to determine parameters for identification of circadian misalignment by visual assessment and developing a machine learning. Discovery of phenodeviance was achieved by screening datasets from 750 mutant lines against primary criteria (mean data) and secondary criteria (at least 50% similar phenotypes, effect size > 1.2 and p-value < 0.001). Validation was conducted by further light entrainment experiments using mutant mice. WTSI: Wellcome Trust Sanger Institute; ICS: Institut Clinique de la Souris; TCP: The Centre for Phenogenomics; RBRC: Riken Bio-Resource Center; HMGU: German Mouse Clinic (Helmholtz Zentrum München). The light schedules in the procedure room were labelled as indicated.

By optimizing conditions to draw scatter plot maps, we further assessed the activity and food intake parameters using 2201 activity/rest scatter plots and 2160 food intake scatter plots at 1-hour intervals over 21 hours (ZT0: light on, ZT12: light off, plots deposited in the Cam-Su GRC database, http://gofile.me/2F1pE/RP2URKxV2) (S1 and S2 Files). The onset times and peak phases for activity and food intake were manually evaluated by observing the curve of

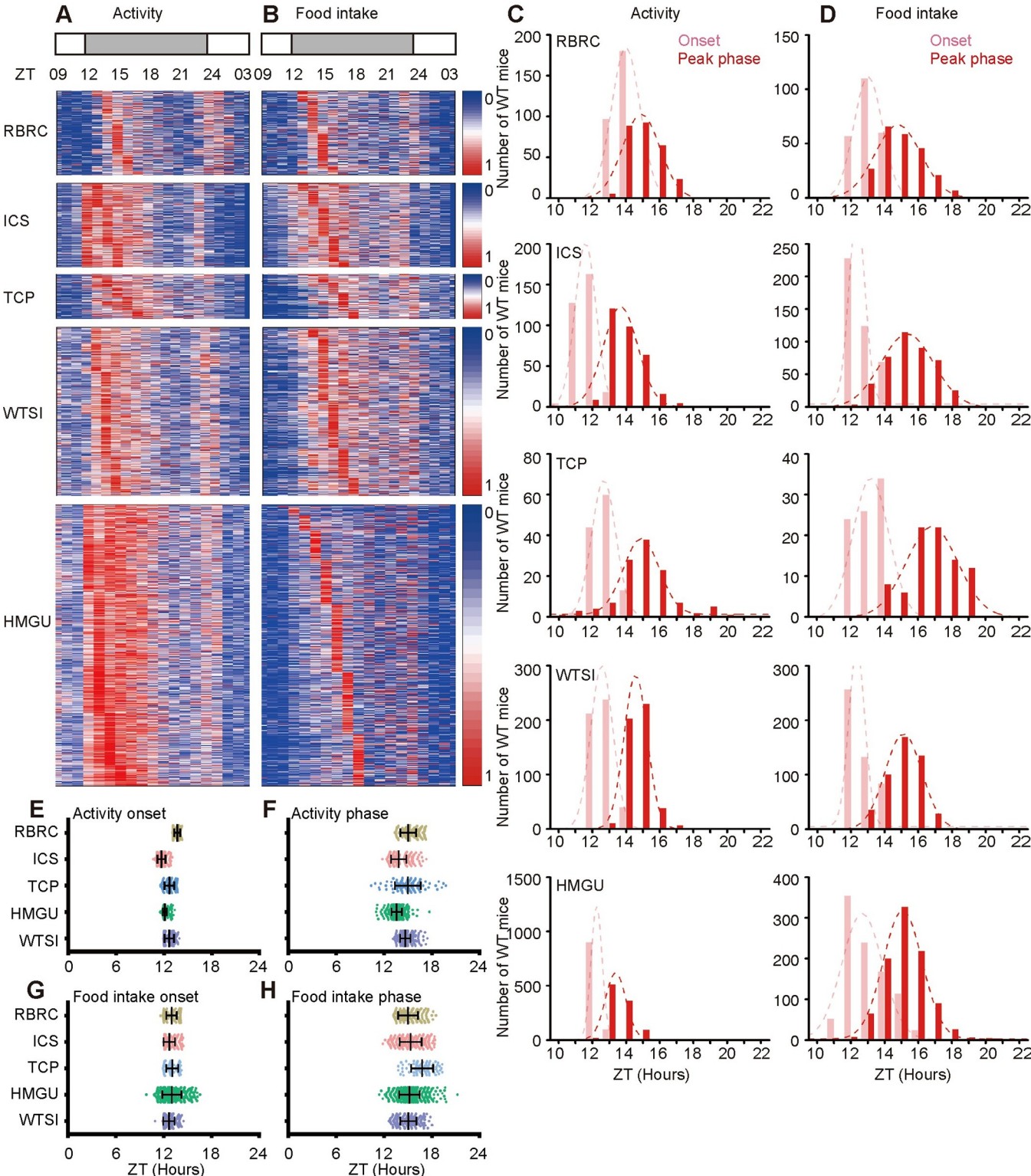

**Fig 2. The distribution of the onset time and peak phase of activity and food intake from IC data from five IMPC centres.** (A, B) Heat map showing the consistent trend of the onset, peak phase and amplitude of activity (A) and food intake (B) recorded by IC of more than 2000 C57BL/6N mice across the five IMPC centres. Mice were ordered according to their evening peak phases. 1 (red) represents the strongest; 0 (blue) represents the weakest. Zeitgeber time (ZT), ZT 0: light on, ZT12: light off. (C, D) The distribution of the onset and peak phase of activity (C) and food intake (D) from the five centres identified by SyncScreener (pink: onset; red: peak phase). Also see S1 Fig with visual assessment. (E-H) The varied range of onset and peak phase of activity and food intake from the five centres identified by SyncScreener.

activity or food intake. We found that the onset time points of activity and food intake were reliably detected with the lowest variance as compared to other parameters from the five different IMPC centres (S1 Fig, pink column, S1 Table). E peak phases of activity/food intake showed a broader distribution than that of the activity onset (S1 Fig, red column, S2 Table). Thus, the onset time and E peak phase are reliable parameters for comparison of circadian misalignment. In addition, because light schedules in the procedure rooms are different in the five centres (light cycle, WTSI: 7:30–19:30; TCP: 7:00–19:00; ICS: 7:00–19:00; RBRC 11:00–23:00; HMGU: 6:00–18:00), our data analyses were restricted to within-centre comparisons instead of between-centre comparisons. We also limited IC data analysis to male mice because the variability of these parameters in females was great, likely due to the estrous cycles.

Empirical visual assessment for onset time and peak phase is time consuming, technically demanding and subjective. To resolve these issues, we employed machine learning to develop predictive models to identify the onset time and peak phase from IC datasets. In our algorithm, a convolutional neural network (CNN) was used to learn from synthetic rhythmic data sets to generate predictive models to investigate the utility of IMPC resources for large-scale screening. Our algorithm needs to learn from labelled training data to build predictive models. Since the IC data are unlabelled and not very diverse, we generated synthetic training data by simulating the pattern of raw data separately in five different centres with labelled onset times and peak phases. The onset time was defined as a transition from rest to active state, like the interim of a piecewise function, while the peak phase was the end of the transition (S2 Fig). Then, we simulated the data before the peak phase by an Ordinary Differential Equation (ODE) with a piecewise function and fitted the remaining data by Gaussian functions. Next, we produced various rhythmic patterns by random disturbance of the onset and peak phase. In addition, measurement noise was modelled as white noise that was added to synthetic training data. Predictive models were generated by learning these large synthetic training data sets. Furthermore, we applied predictive models to the IC datasets from 2201 mice for activity/rest and 2160 mice for food intake in five different centres. The results for the onset time and peak phase of activity and food intake for each centre were determined by SyncScreener (Fig 2C–2H, S3 and S4 Tables).

Next, we compared the results from visual assessment and machine learning algorithm (SyncScreener, S3 File) using Bland–Altman plots to determine whether the automated parameter determination was accurate compared to visual assessment [28]. Bland–Altman plots showed that the differences between the two methods were acceptable within a 95% limit of agreement with more acceptable results for the onset time than the peak phase (S3 Fig and S4 Fig).

Finally, since we did not find known clock regulators with activity phenotypes from screened mutants, we used several known circadian mutant lines to validate our predictive models and parameters: $hPER2^{S662G}$ mice (advanced onset of activity), $Fbxl3^{-/-}$ mice (delayed onset of activity), $Nestin$-$Cre$;$Zbtb20^{-/-}$ mice (delayed peak phase) and $hPER1^{S714G}$ mice (advanced onset of food intake) [15,16,29,30]. These clock mutant mice were placed in calorimetry cages to generate IC data plotted at 1-hour intervals over 24 hours (Fig 3A–3J). The onset and peak phase were estimated by SyncScreener (Fig 3K) and visual assessment for comparison (S5 Table). The standardized effect size ($d$) was used to estimate the phenodeviance where the absolute difference between the mutant and wild-type control was scaled in units of the phenotypic standard deviation (a statistical power analysis according to IMPC protocol) [31]. The $p$ value was employed to measure a false-positive risk. As shown by the SyncScreener results in Fig 3K, these known circadian mutants showed the estimated detectable $d$ value (effect size: 1.2–6.64) and clearly statistically significance ($p < 0.05$–$0.001$) for the onset time and peak phase that were consistent with their known phenotypes (Fig 3K). These results

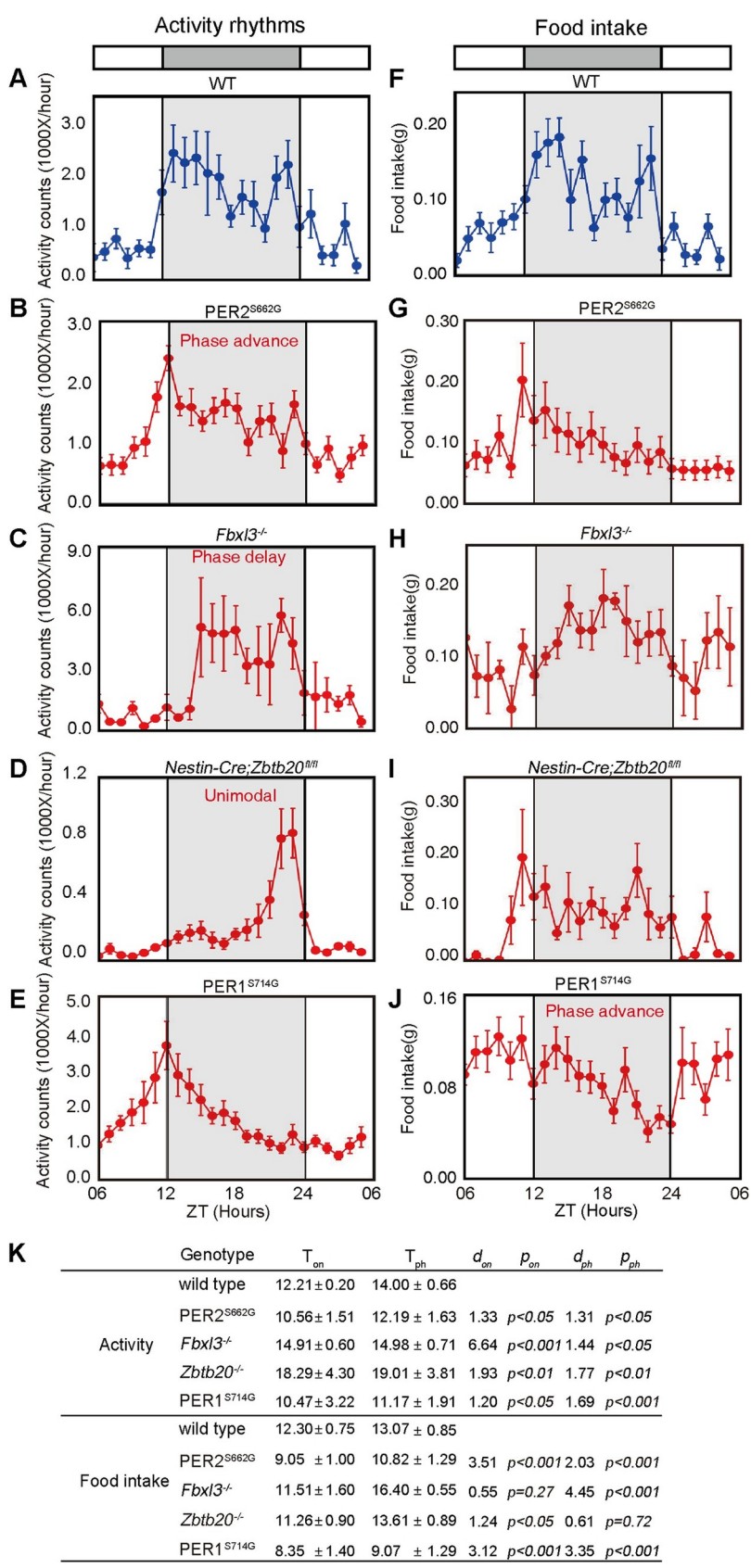

**Fig 3. Established positive control for circadian misalignment of activity and food intake behaviour.** (A-J) Recording of activity (A-E) and food intake (F-J) in known mutant lines and analyses of onset and peak phase by CLAMS. Data represent the mean ± SEM measured by IC under light-dark cycles from 8–14 mutant male mice for the indicated genotype. (K) The onset time ($T_{on}$) and peak phase ($T_{ph}$) identified by SyncScreener. $d$: detectable value (effect size) for the onset ($d_{on}$) and peak phase ($d_{ph}$); $p$: statistical significance by t-test for the onset ($p_{on}$) and peak phase ($p_{ph}$). Also see S5 Table for visual assessment results.

suggest that determination of these two parameters by our machine learning algorithm and IC data are reliable for identification of mouse mutants with impaired circadian misalignment behaviour.

## Identification of mouse mutants with impaired circadian misalignment behaviour

*Primary criteria*: We applied the machine learning algorithm to detect the onset and peak phase of activity/food intake in mutant mouse lines. By comparing the parameters of the wild-type control mice from each centre, we screened 440 homozygous and 310 heterozygote mutant strains, representing loss-of-function of 726 unique genes (S6 and S7 Tables). We obtained a mean scatter dot curve generated from 7–8 male mice for each genotype available in the IMPC database to evaluate the onset times and peaks by SyncScreener. The distribution of phase deviations of onset and peak phases between the wild-type and mutant mouse lines at each centre are shown in Fig 4A–4E and S8 Table. The onset time or peak phase which falls in the tails beyond ~+/- 2 s.d. from mean (approximate 5%) were designated outliers, where s.d. represents the standard deviation of differences between the mutants and wild-type control in each centre. Among the 750 mutant lines, 12 existed at two or more centres and they all exhibited the same phenotypes as the wild-type (S9 Table). 88 (11.7%) of the 750 mutant mouse lines falls in the tail beyond ~+/- 2 s.d (approximate 5%) from mean for either the onset and/ or peak phase of activity and food intake. Those mutant lines were selected against the secondary criteria (S10 Table, 88 genes).

*Secondary criteria*: The candidate mouse lines were further examined by the following criteria: (1) 50% of individual mice within a line displayed similar phenotypes as one of phenotyping baseline as described previously [24,32,33], (2) standardized effect size (Cohen's $d$), where $d = \frac{\overline{x_m} - \overline{x_{wt}}}{s}$ is the absolute difference between mutant and baseline means scaled in units of phenotypic standard deviation. A larger value of effect size always indicates a stronger phenodeviance [24,32]. Effect size ($d$) >1.2 suggested that the group difference is large between the mutant mice and the wild-type mice, and (3) the two-tailed t-test was used to examine the statistical significance ($p$ value), where $p<0.001$ indicated statistical significance between the mutant line and phenotyping baseline. Five mutant lines ($Slc7a11^{tm1b/tm1b}$, $Rhbdl1^{+/tm1.1}$, $Spop^{+/tm1b}$, $Oxtr^{tm1.1/tm1.1}$, $Ctc1^{+/tm1b}$) met the above three criteria (50% mice with similar phenotype, $d > 1.2$, and $p < 0.001$) (Fig 5 and S11 Table). For the $Slc7a11^{tm1b/tm1b}$ mice, the time of activity onset is advanced compared to the wild-type at ICS (from ZT8 to ZT11) and the phase of food intake is indistinguishable between mutant and wild-type mice (Fig 5A and 5B). $Rhbdl1^{+/tm1.1}$ mice displayed a delayed onset of activity compared to wild-type mice at TCP (Fig 5C and 5D). Onset of activity and food intake in $Spop^{+/tm1b}$ mice were delayed (Fig 5E and 5F). The $Oxtr^{tm1.1/tm1.1}$ mice exhibited a trend towards more daytime activity tendency than wild-type mice at the RBRC (Fig 5G and 5H, S5 Fig). The $Ctc1^{+/tm1b}$ mice showed a delayed onset of both activity and food intake (Fig 5I and 5J). The effective size and $p$ value from all candidate mutants fall within positive ranges (S11 Table, compared to Fig 3K for positive controls) and these mutant mice were labelled as the outliers in the Fig 4. These results suggest

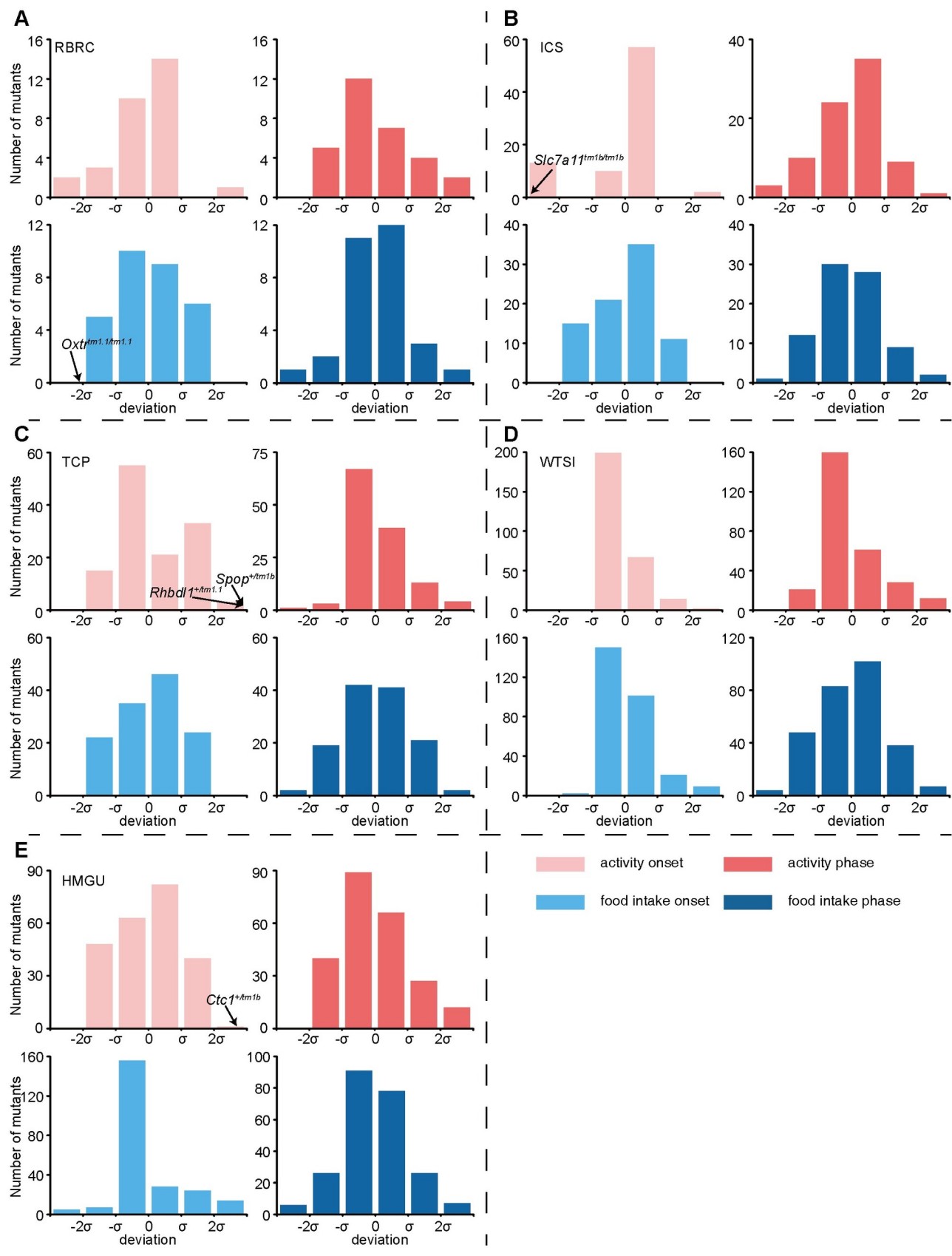

**Fig 4. Systematic identification of the onset and peak phase phenotypes.** (A-E) Distribution of the number of mutant lines with onset or peak phase deviated from wild-type mice at ±0, ±σ, and ±2σ. IC data from RBRC (A), ICS (B), TCP (C), WTSI (D) and HMGU (E) are presented separately. Mutants were defined as outliers when the parameters deviated from the mean greater than 2σ. Five candidates meeting the secondary criteria are labelled by arrows.

that these five genes are potentially involved in circadian misalignment. The *Spop*, *Ctc1* and *Rhbdl1* homozygous mice were preweaning or embryonic lethality, so the phenotypes of homozygous knockout is not known. In addition, we found that the *Slc7a11*$^{tm1b/tm1b}$, and *Rhbdl1*$^{+/tm1.1}$ mutant mice showed altered glucose tolerance (IMPC data, S12 Table), suggesting that the deletion or haploinsufficiency of these genes impaired metabolism.

## Confirmation of the role of *Slc7a11* in circadian behaviour

To confirm our screening results from the IMPC IC datasets, we generated the *Slc7a11*$^{tm1a/tm1a}$ mice using targeted C57BL/6N ES cells by Cam-Su Genomic Research Center [34] (S6 Fig). Consistent with that of the IMPC results with advanced onset of activity in Fig 5A, *Slc7a11*$^{tm1a/tm1a}$ mice exhibited advanced onset of activity on the first and third day under LD cycles by the Comprehensive Lab Animal Monitoring System (CLAMS, IC) (Fig 6A and 6B). Although the onset was comparable between the *Slc7a11*$^{tm1a/tm1a}$ and wild-type littermates in the second day, the mutant mice showed a phase advance for the declining activity phase for the next dawn (Fig 6A, red arrow). Consistent with the results of SyncScreener, the food intake was comparable between mutant and wild-type mice, including VO$_2$, VCO$_2$ (S7 Fig). We next analyzed voluntary wheel-running activity to evaluate the onset time, and free-running period of the circadian clock for the *Slc7a11*$^{tm1a/tm1a}$ mice. Under LD cycles, the activity onset times were significantly advanced and unstable with a larger variance in the *Slc7a11*$^{tm1a/tm1a}$ mice compared with wild-type littermates (Fig 6C and 6D). This result is consistent with the above observations, suggesting that the mutant mice have reduced sensitivity to photic entrainment. The mice were subsequently released to constant darkness (DD) to determine circadian period. Of note, there is no significant difference in the circadian period (Fig 6E–6G). To fully examine the role of *Slc7a11* in circadian entrainment, we subjected *Slc7a11*$^{tm1a/tm1a}$ mice and wild-type littermates to a simulated jet-lag environment. In response to a 6 h advance shift of the LD cycle, *Slc7a11*$^{tm1a/tm1a}$ mice immediately showed phase advance (S8 Fig). Furthermore, to minimize the masking effect of light, *Slc7a11*$^{tm1a/tm1a}$ mice were subjected to a skeleton photoperiod with 15-min light arms from clock time 07:45 to 08:00 and from clock time 20:00 to 20:15 (ZT0, clock time 08:00) with darkness at all other times. We found that the activity onset was significantly advanced in *Slc7a11*$^{tm1a/tm1a}$ mice compared with wild-type littermates (Fig 6H–6J). Furthermore, mice were exposed to an alternating cycle of 3.5 h light and 3.5 h dark for T-cycle experiments. *Slc7a11*$^{tm1a/tm1a}$ confined their activity mostly to the dark phase as did their littermates (S9 Fig), suggesting that disruption of *Slc7a11* did not affect masking behaviour. Altogether, these data suggested that disruption of *Slc7a11* changes sensitivity to circadian entrainment. Since the suprachiasmatic nucleus (SCN) is responsible for entraining mice activity rhythm by light/dark cycles [35], we examined the expression of *Slc7a11* in the SCN by *in situ* hybridization at ZT6 (day time) and ZT18 (night time). We found that *Slc7a11* hybridization signals were higher at ZT 18 than those at ZT6, suggesting cyclic expression of *Slc7a11* mRNA under LD cycle (Fig 7A and 7B). In addition, we also examined the expression profiles of *Slc7a11* in the wild-type mice under constant darkness by RT-PCR [36,37]. The profiles of *Slc7a11* expression showed rhythmicity in the wild-type SCN by Jonckheere-Terpstra-Kendall (JTK) cycle analysis (Fig 7C, p<0.05). In *Slc7a11*$^{tm1a/tm1a}$ mice, we found that the mRNA levels of most core clock genes were comparable to wild-type mice in the SCN and

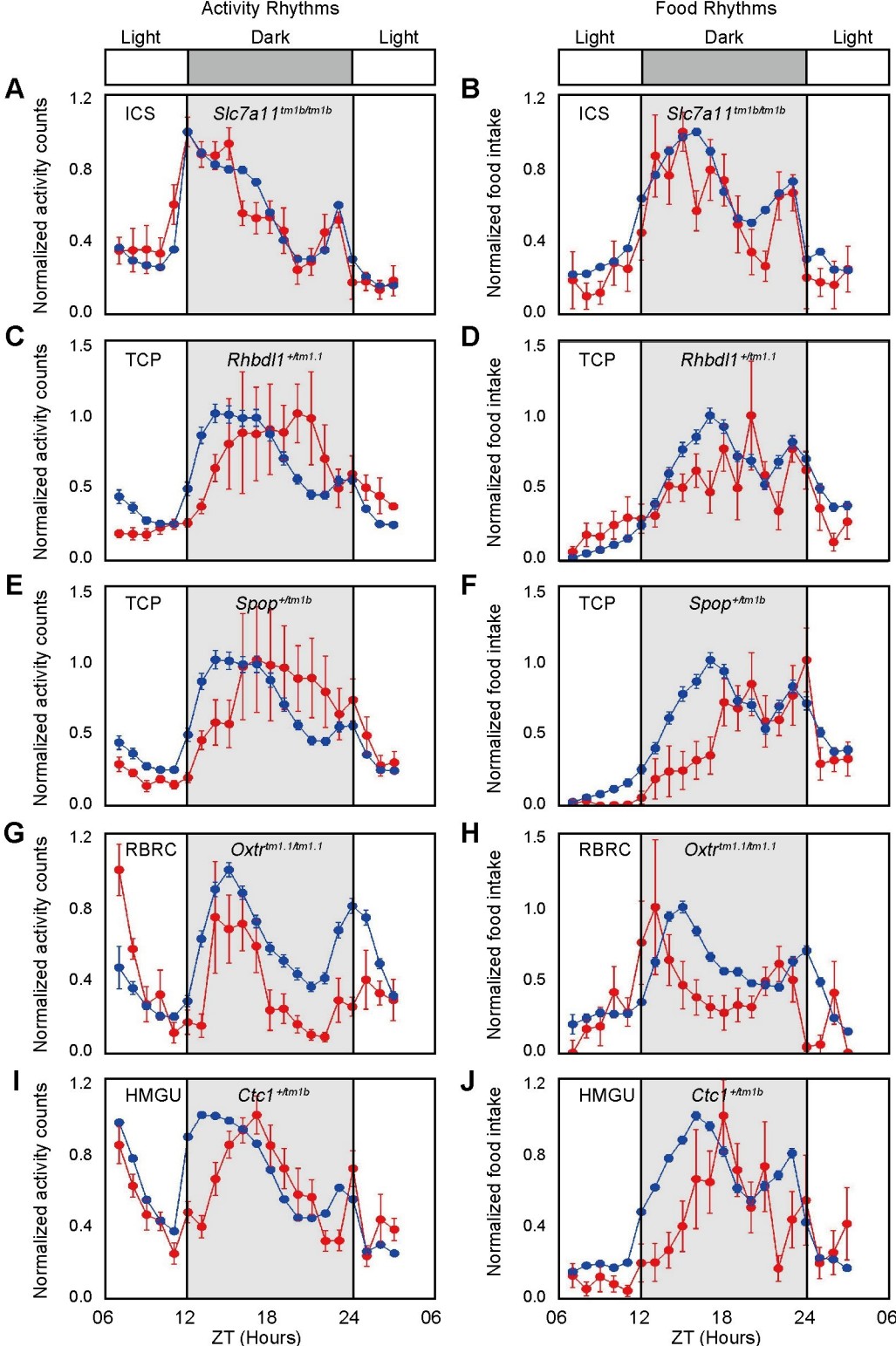

**Fig 5. Analysis of onset and peak phase phenotypes of mice from identified mutant lines.** Profiles of oscillating activity (A, C, E, G, I) and food intake over time (B, D, F, H, J) for *Slc7a11* (A, B), *Rhbdl1* (C, D), *Spop* (E, F), *Oxtr* (G, H) and *Ctc1* (I, J) mutant mice. Blue and red lines represent the wild-type and mutant mice in the same centre, respectively. Data are normalized and presented as the mean value ± SEM (n = 7–8 for each genotype). The time of day is indicated in hours, and the dark period is indicated by shading.

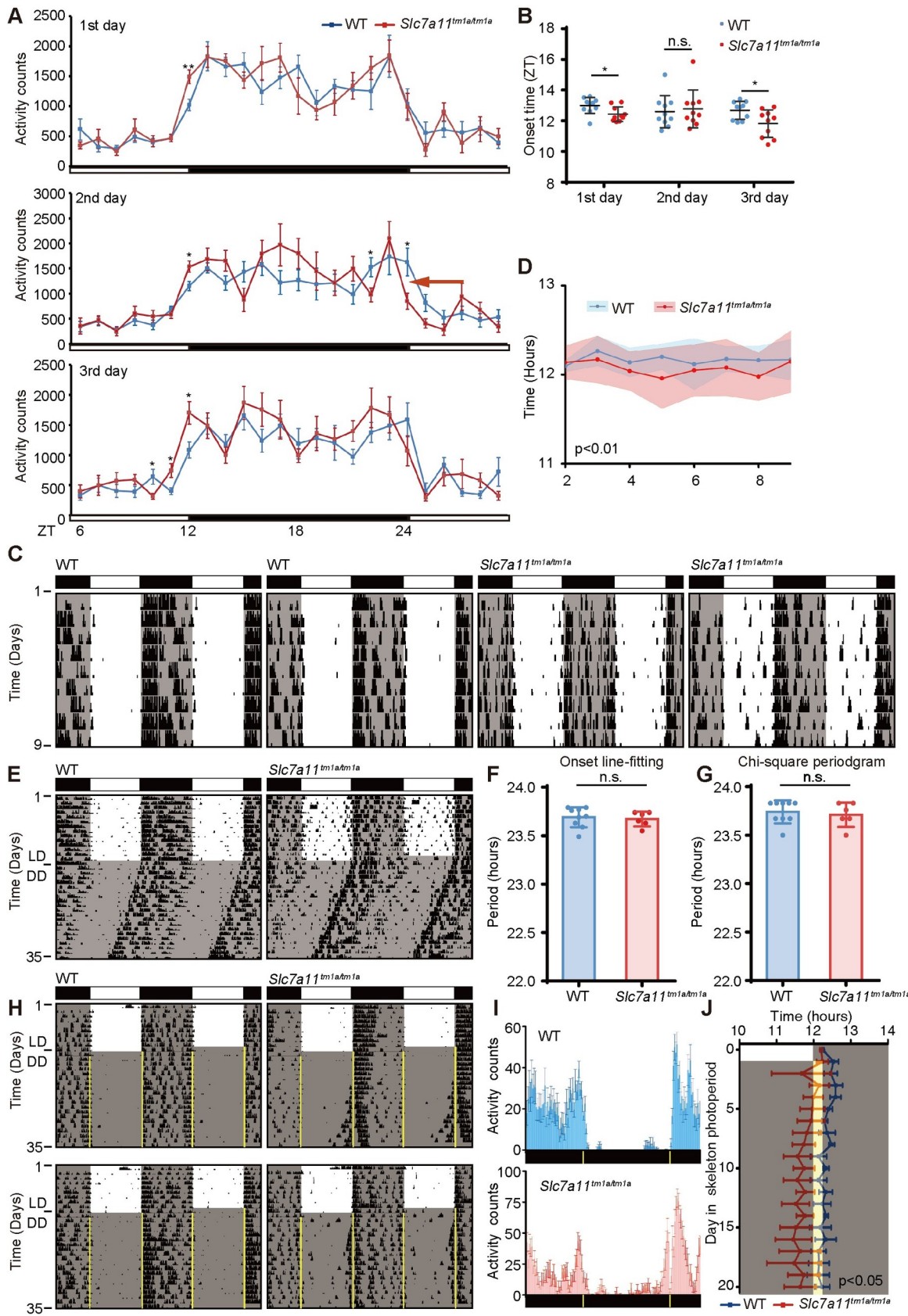

**Fig 6. Validation of the role of *Slc7a11* in circadian entrainment of activity onset.** (A) The activity profiles of *Slc7a11*$^{tm1a/tm1a}$ and wildtype littermates under a 12hr light/12hr dark cycle using CLAMS (IC data). Rhythms were plotted over a 24 hr time frame as the mean ±; SEM for three days (n = 10 for each genotype). (B) The onset of activity was shown as mean ± SEM. The Student t-test is used to determine the significance, *p < 0.05, **p < 0.01. (C) The representative actograms for analyzing the onset times under LD cycles in wild-type mice (left) and *Slc7a11*$^{tm1a/tm1a}$ mice (right). (D) The onset times of activity were measured by Clocklab analysis. Data was shown as means with SEM (n = 8 for WT, n = 6 for *Slc7a11*$^{tm1a/tm1a}$) for 8 continuous days. The pink and blue shadow indicates the range of onset times for the *Slc7a11*$^{tm1a/tm1a}$ mice and their wild-type littermates. (E) The representative actograms of wheel-running activity for period determination. The mice were first entrained to an LD cycle for 14 days and then released in DD for approximately 3 weeks. Black shading indicates the time when lights were off, and the white box indicates the time when lights were on. (F-G) Period was determined by line fitting of activity onset (F) and chi-square periodogram (G) from day 11 to day 21 in DD. (H-I) Representative actograms (H) and activity profiles (I) of wheel-running activity for wild-type mice and *Slc7a11*$^{tm1a/tm1a}$ mice under skeleton photoperiod with 15-min entraining arms for an 20-d period (n = 6 for each genotype). Each row represents a single day. The mice were first entrained to a LD cycle for 10 days and then released to the skeleton photoperiod with 15-min light arms from clock time 07:45 to 08:00 and again from clock time 20:00 to 20:15 (ZT0, clock time 08:00). The black and yellow bars in I (below) represent periods of darkness and light, respectively. (J) The onset times of activity under skeleton photoperiod were measured by Clocklab analysis. Blue: WT; Red: *Slc7a11*$^{tm1a/tm1a}$. Data was shown as means with SEM. Two-way ANOVA was employed to analyze statistical significance.

liver tissues (Fig 7D and S10 Fig). This is consistent with the non-altered circadian period. However, the expression phases of genes with high-amplitude such as *Per2* and *Dbp* were advanced in the SCN of *Slc7a11*$^{tm1a/tm1a}$ mice compared with wild-type littermates (Fig 7D), in consistent with the results of advanced activity onset, suggesting that disruption of *Slc7a11* affects the circadian entrainment. These phase advances are not found in the *Slc7a11*$^{tm1a/tm1a}$ liver tissues (S10 Fig). In addition, the SCN mRNA levels of *Grp*, *Grpr*, *Vip*, *Pk2* and *Pkr2* were significantly altered at some time points in DD in *Slc7a11* knockout mice (Fig 7E). The profiles of GRP and GRPR are rhythmic under LD with a peak at ZT12 in the SCN that regulate the circadian phase [38,39]. The misalignment between the expression of *Grp* and *Grpr* observed in the SCN of *Slc7a11* knockout mice may result in missing the best binding timing and result in a defect in entrainment. In addition, VIP participates the synchrony in mammalian clock neurons and mediates the entrainment of circadian rhythms [8,40]. The plateau of *Vip* expression in the SCN from *Slc7a11*$^{tm1a/tm1a}$ mice instead of the sharp peak of *Vip* expression at ZT 12, as well as the phase shift of *Avp*, appears to weaken the optimal onset of activity (Fig 7E). The function of prokineticin 2 (*Pk2*) and its receptor *Pkr2* have been shown to be under the dual regulation of both light and the circadian clock and affect the circadian entrainment [41,42]. The altered expression of these genes involved in SCN intercellular communication suggest that the SCN neuron synchronization might be affected in the mutant mice [43]. Together, our results suggest that although *Slc7a11* is not required for core clock function, it is involved in mediating circadian entrainment of behaviour.

## Discussion

In this study, we demonstrated the feasibility of large-scale characterization of mouse mutants with impaired circadian alignment under light/dark cycles as part of an IMPC collaborative effort to generate a genome-wide catalogue of gene function. By analyzing IC datasets of 2201 C57BL/6N mice for activity/rest and 2160 C57BL/6N mice for food intake from five IMPC centres, we identified two robust parameters of circadian misalignment of behaviours. Our machine learning based SyncScreener enables fast, objective and large-scale behavioural screening of mutant mouse lines. We identified five genes (approximately 0.66%) among 750 mutant lines that are potentially involved with circadian misalignment of activity and food intake behaviour. Previous studies have mainly focused on measuring circadian period and not misalignment as the target phenotype. Because IMPC plans to generate ~20,000 mutant lines, many for genes thus far uncharacterized, our results have laid the foundation for future a comprehensive screen of circadian behaviour mutants under light/dark cycles.

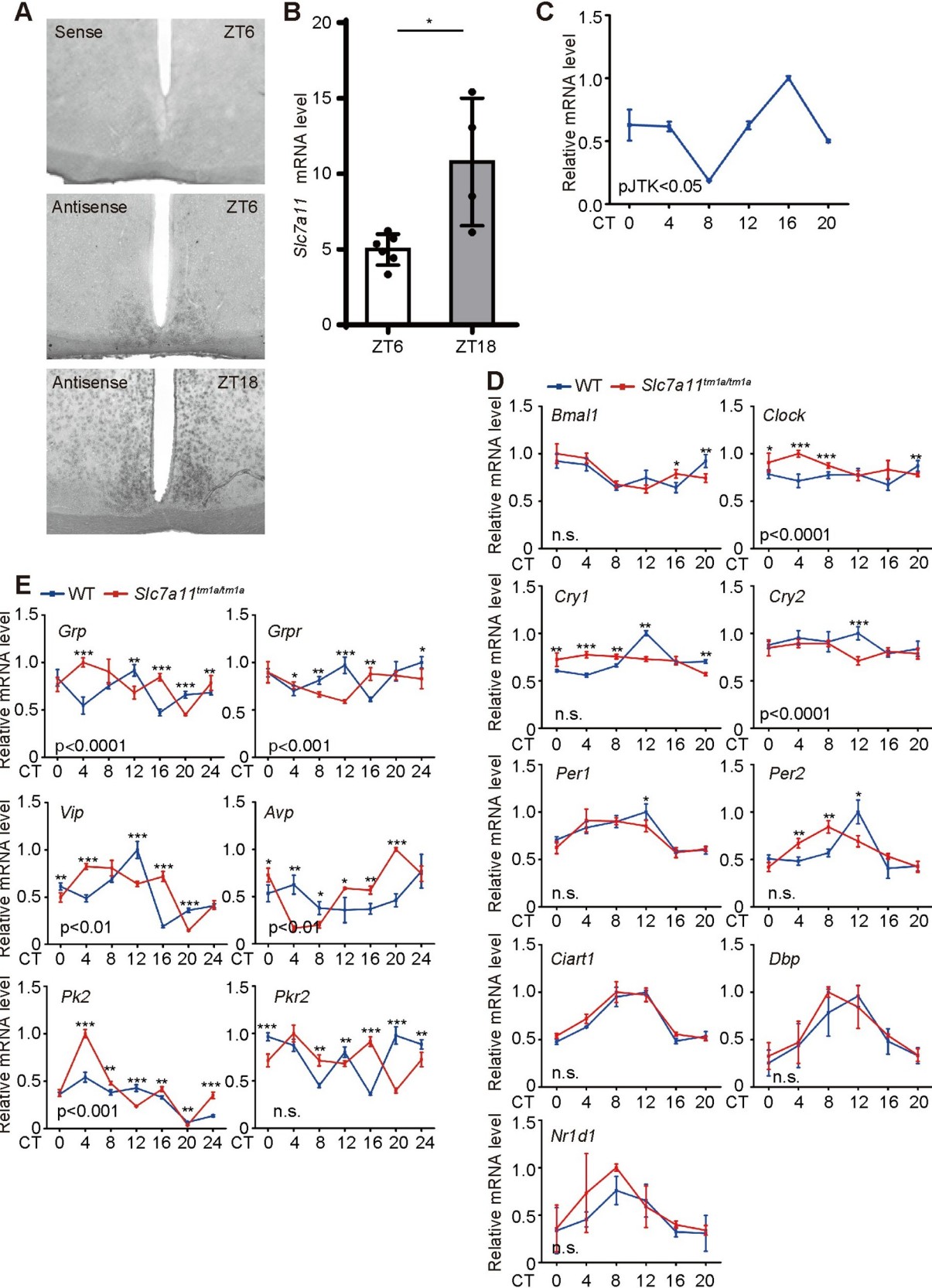

**Fig 7. Altered expression of the coupling genes in the *Slc7a11*^*tm1a/tm1a* mice.** (A) Expression of *Slc7a11* in mouse SCN detected by *in situ* hybridization at ZT6 and ZT18. ZT, Zeitgeber Time. Coronal brain sections containing the SCN were hybridized with the cRNA sense (upper) or antisense probe (middle and lower) of *Slc7a11* at ZT6 and ZT18. (B) Quantification of *in situ* hybridization signal of *Slc7a11* by Image J from 3–4 coronal brain sections. *: p < 0.05. (C) Real-time PCR analysis of the expression of *Slc7a11* in SCN of wild-type mice. Error bars represent the s.d. for each time point from three biological independent replicates. The rhythmicity of gene expression was determined based on the JTK algorithm (pJTK < 0.05). (D) Expression profiles of the core clock genes in the SCN from control and *Slc7a11*^*tm1a/tm1a* mice. Also see the expression in the liver (S10 Fig). Error bars represent the s.d. for each time point from three independent replicates. (E) Expression profiles of the coupling factors in the SCN from control and *Slc7a11*^*tm1a/tm1a* mice. Error bars represent the s.d. for each time point from three independent replicates. Two-way ANOVA was employed to test the statistical significance. *: P < 0.05; **: P < 0.01; ***: P <0.001.

Anticipating and adapting to light/dark cycles is a major function of circadian clocks. Recent discoveries have highlighted how the internal coincidence of the circadian clock can change phases to synchronize with external environmental cycles[44], as has been shown by mutations of *PER2* and *CSN1KD* in familial advanced sleep phase syndrome[14,15], mutations or SNPs of *ARNTL*, *DEC1* and *RORB* in bipolar disorder[45], and mutations of *CRY1* in familial delayed sleep phase disorder[12]. The identification of these genes provides important insights into how molecular clocks affect human health and behaviour. In this study, we discovered that *Slc7a11*, *Spop*, *Rhbdl1*, *Oxtr* and *Ctc1* are potential candidate genes that are involved in the precision and adaptability of circadian behaviours under light-dark cycles. Validation of one of these candidate genes, by generating *Slc7a11* mutant mice in Cam-Su Genomic Resource Center, revealed that *Slc7a11* is involved in interfering intercellular coupling factors in the SCN. Further studies of these candidate genes will uncover new insights into the mechanism of circadian misalignment of behaviours. A critical future next step will be to determine how these genes, which may be involved in distinct pathways, can influence the phase of behaviour and physiology in response to light/dark cycles. Ultimately, the availability of more than 20,000 mouse lines and our screening method established here should allow a comprehensive identification of genes involved in misalignment of mouse behaviour under light/dark cycles.

## Materials and methods

### Ethics statement

Mouse studies were approved by the Animal Care and Use Committee of the CAM-SU Genomic Resource Center (CAM-SU-AP#: YX-2017-1), The Centre for Phenogenomics (TCP) (Approval committee: Animal Care Committee (ACC) of The Centre for Phenogenomics. Approval License: Animal Use Protocol (AUP) 0275 and 0279H), GMC Helmholtz Zentrum München (HMGU) (Approval License: 144–10), ICS Mouse Clinical Institute (ICS) (Approval Committee: Com'Eth N˚17 and French Ministry for Superior Education and Research (MESR). Approval licenses: MESR: APAFIS#4789–2016040511578546), RBRC RIKEN Tsukuba Institute, BioResource Center (RBRC) (Approval License: Exp11-002, 12–002, 13–002, 14–002, 15–002, 16–002 Collection, maintenance, storage, breeding and distribution of the mouse resources Exp11-011, 12–011, 13–011, 14–009, 14–017, 15–009, 16–008 Phenotyping analyses and related studies in mice), WTSI Wellcome Trust Sanger Institute (WTSI) (Approval License: PPL 80/2076 Valid 27th Nov 2006 - 3rd Jan 2012; PPL 80/2485 valid 3rd Jan 2012 - 5th Dec 2016). Every effort was made to minimize the number of animals used, and their suffering.

### Animals

Mice were housed in specific pathogen-free animal facilities. *Slc7a1*^*1tm1a/tam1a* mice were generated according to the standard protocol in CAM_SU Genomic Resource Center. *Fbxl3*,

*Zbtb20*, *hPER1*[S730G] mutant mice were generated as described previously and *hPER2*[S662G] was generated by Dr. Xu in Fu & Ptacek lab in the UCSF. IC data from these known mutant mice were generated by Xu lab in the CAM-SU Genomic Resource Center. All mutant mice used in these studies have been described previously [15,16,29,30].

## Phenotype data acquisition

Indirect Calorimetry raw data was collected from five centres (WTSI, TCP, ICS, RBRC and HMGU). Five centres follow the pipeline of IMPC. IC equipment: WTSI, ICS and HMGU: TSE PhenoMaster/Labmaster CaloSys, TCP: Columbus Oxymax/CLAMS, RBRC: O'hara FWI-3002 & IA-16M, CAM-SU: Oxymax/CLAMS. Light schedules are WTSI: 7:30–19:30; TCP: 7:00–19:00; ICS: 7:00–19:00; RBRC 11:00–23:00; HMGU: 6:00–18:00. Recording in each center started and ended at different times. In the indirect calorimetry module standard measurements begin five hours before lights-off (lights off = T0) and are finished at T16 i.e. four hours after lights-on the next morning. Data analysis were restricted to within-centre comparisons between controls and mutants.

## Data pre-processing

We improved the accuracy of the automatic identification of the peak phase and onset time by detecting and removing the bad data points (measurement errors and environment interferences) in raw data for activity and food intake before running any analyses. First, we excluded out-of-range values that exceeded a realistic scope, such as a food intake of more than 0.65 g and an activity higher than 4000 counts per hour during the daytime. Second, we carefully assessed the data obtained at the light-off time point ZT12 (denoted as $y_{12}$), particularly the data obtained by ICS, because abnormal data generated at ZT12 resulting from an unstable environment, such as unstable light intensity, may lead to an inaccurate assessment of the onset time and peak phase. If $y_{12}$ satisfied the following conditions: 1) a local maximum, 2) higher than four-fifths of the peak value (denoted as $y_{max}$), and 3) no other local maximum between ZT12 and ZT17, we considered it a bad data point and removed it from the dataset. Finally, we deleted the pulse breakup data point that was far from the neighbouring data points. In brief, a data point can be regarded as breakup data when the difference, $y_{diff\_\tau} = \min \{|y_\tau - y_{\tau-1}|, |y_\tau - y_{\tau+1}|\}$, reaches a certain threshold, $y_{th\_p} = \alpha_1 y_{max}$, where $y_\tau$ represents the data value corresponding to $ZT_\tau$, and the parameter $\alpha_1 = 0.57$ is chosen empirically.

## Onset time and peak phase recognition by deep learning (CNNs)

We applied a machine learning algorithm to identify the onset time and peak phase of daily cyclic data. In our algorithm, a CNN learned synthetic rhythmic data sets was used to predict the two biological parameters.

## Training data set

The inputs to a machine learning algorithm are thousands of labelled measurements of samples that are the same type as measurements used to predict. Here, however, the diversity of chorotypes in our data was poor, the rhythm of most tested mice was similar to that of wild-type, and the rhythm of most volunteers was normal. Furthermore, our work was to label (onset times and peak phases) the daily rhythmic mice activity/food intake data. Therefore, we generated training data sets according to the raw activity and food intake data from five centres. We were interested in the onset times and peak phases of mouse behaviour. The onset time is the first transition point at which the state of mouse behaviour transfers from the rest

state to the active state, and the transition process ends at the peak phase. As the transition process is kinetic, we simulated the data before the first peak by an ODE and fitted other data by Gaussian functions. Finally, we employed white noise to simulate measurement errors. To produce proper training data, we performed the following steps:

First, we calculated the average values ($y_{ave,i}$, i = 1,2,3...25) of all measurements at 25 zeitgeber times.

Second, we simulated the state transition process (average data before the first peak) via an OED with a piecewise function. ODE can be described as follows:

$$\frac{dx}{dt} = \varepsilon(F - x)$$

$$F(t) = \begin{cases} \dfrac{u_2 h_2 - u_1 h_1}{t_2 - t_1} t + \dfrac{u_1 h_1 t_2 - u_2 h_2 t_1}{t_2 - t_1} & t \leq t_2 \\ u_3 h_3 & t_2 < t < t_3 \end{cases}$$

$F$ is a piecewise function presenting the endogenous switch where $t_1$, $t_2$ and $t_3$ stand for the zeitgeber times of the first average data point one hour before the onset time and peak phase. $h_2$ and are data values at $t_2$ and $t_3$. $h_1$ is set empirically. $u_1$, $u_2$ and $u_3$ are all set to 1.1. $x$ is the measurements of mouse behaviour, which follow the biological switch $F$ in this model.

Mouse behaviour is bimodal in the day/night cycle. However, in some cases, there are three peaks. Thus, we employed three Gaussian functions for fitting the remaining average data. Gaussian functions $f_1(\overrightarrow{a_1, t})$, $f_2(\overrightarrow{a_2, t})$ and $f_3(\overrightarrow{a_3, t})$ can be described as follows:

$$f_1(\overrightarrow{a, t}) = a_1 e^{-\frac{(t-a_3)^2}{2a_2^2}} + a_4$$

$$f_2(\overrightarrow{a, t}) = a_5 e^{-\frac{(t-a_7)^2}{2a_6^2}} + a_8$$

$$f_3(\overrightarrow{a, t}) = a_9 e^{-\frac{(t-a_{11})^2}{2a_{10}^2}} + a_{12}$$

where $a_3$, $a_7$ and $a_{11}$ represent peak phases of three peaks; three terms $a_1$, $a_5$ and $a_9$ stand for the amplitudes of three peaks; $a_2$, $a_6$ and $a_{10}$ describe the width of each peak; and $a_4$, $a_8$ and $a_{12}$ are the minima of the three fitting curves. Then, we found the initial values and parameter ranges from raw data for Gaussian fitting. We detected the local maximums that were higher than four neighbouring data points. The corresponding ZT values of local maximums ($ZT_{peak1}$, $ZT_{peak2}$ and $ZT_{peak3}$) were used as initial values for $a_3$, $a_7$ and $a_{11}$. We took the initial values of $a_2$, $a_6$ and $a_{10}$ as 2 empirically. The initial $a_5$ and $a_9$ are $m_2 - min(y_{ave})$ and $m_3 - min(y_{ave})$ where $m_2$ and $m_3$ are the measured peak values of the second and third peaks from raw data, and the initial $a_1$ is $max(x_{onset}) - min(x_{onset})$. The initial values of $a_4$, $a_8$ and $a_{12}$ are 0.05 for food intake, 750 for activity, respectively. Then, we set the lower and upper bounds for the above parameters. The ranges of $a_7$ and $a_{11}$ are $[ZT_{peak2},-3, ZT_{peak2}+3]$ and $[ZT_{peak3},-3, ZT_{peak3}+3]$, and the ranges of $a_3$ are $[ZT_{peak1}, ZT_{peak1}]$. The ranges of $a_5$ and $a_9$ are $[0,10(m_2 - min(y_{ave}))]$ and $[0,10(m_3 - min(y_{ave}))]$, and the ranges of $a_1$ are $[(max(x_{onset}) - min(x_{onset})),(max(x_{onset}) - min(x_{onset}))]$. The ranges of $a_2$, $a_6$ and $a_{10}$ are all $[0, 4]$. The ranges of $a_4$, $a_8$ and $a_{12}$ are $[0, 0.15]$ for food intake. If there are only two peaks, $m_3$ is set to zero. In the end, raw average data after the first peak were fitted with multiple Gaussian functions via MATLAB's lsqcurvefit function with above initial values and parameter ranges. Accordingly, we obtained standard parameters $\overrightarrow{p^*}$ to best fit the ODE and Gaussian functions to raw average data.

Next, we generated various parameter sets $\vec{p}$ to mimic different chorotypes. The parameters for ODE can be described as follows:

1. $t_1$ is the zeitgeber time of the first data point. $t_1$ is set to 6 for mice and 0 for humans;

2. $t_2$ is the zeitgeber time of one data point before onset. $t_2$ is uniformly distributed between 9 and 13 for mice, and

3. $t_3$ is the zeitgeber time of the peak phase. $t_3 = t_2 + \Delta t$, $\Delta t$ is uniformly distributed between 0 and 3.

4. $h_3$ is the peak height of the first peak. $h_3 = \zeta_{h3} h_3^*$. $h_3^*$ represents the peak height of the first peak of the standard curve. $\zeta_{h3}$ is a random variable following uniform distribution $U(0.5, 1.5)$.

5. $h_2$ is the height of the data point at $t_2$ $h_2 = \zeta_{h2} h_3$. $\zeta_{h2}$ is a random variable following uniform distribution $U(0.2, 0.9)$.

6. $h_1$ is the height of the data point at $t_1$. $h_1 = h_2 / \zeta_{h1}$. $\zeta_{h1}$ is a random variable following uniform distribution $U(1, 1.2)$.

7. $u_1 = u_2 = u_3 = 1.1$.

8. $\varepsilon = \zeta_\varepsilon \varepsilon^*$. $\zeta_\varepsilon$ is a random variable following uniform distribution $U(0.8, 1.2)$.

9. The solution of ODE with the above parameters is a time series $y_1$, which represents the pattern before the first peak.

Parameters for three Gaussian functions can be described as follows:

1. $a_1$ is the amplitude of the first peak, $a_1 = \max(y_1) - y_1(1)$.

2. $a_2$ is the width of the first peak. $a_2 = \max\{t_{width1}, t_{width2}\}$, $t_{width2}$ is uniformly distributed between 2 and 3. $t_{width1} = \zeta_{width} t_{width0}$, $\zeta_{width}$ is uniformly distributed between 0 and 3. $t_{width0} = t_3 - t_{mid}$, $t_{mid}$ is the zeitgeber time of $y_{mid}$, $y_{mid} = \frac{h_2 + h_3}{2}$.

3. $a_3$ is the zeitgeber time of the first peak phase, $a_3 = t_3$.

4. $a_4$ is the minima of the curves, $a_4 = y_1(1)$.

5. $a_5$ is the amplitude of the second peak, $a_5 = \zeta_{a5} a_1$. $\zeta_{a5}$ is a random variable following uniform distribution $U(0.8, 1.2)$.

6. $a_6$ is the width of the second peak. $a_6 = \zeta_{a6} a_6^*$. $a_6^*$ represents the width of the second peak of the standard curve, and $\zeta_{a6}$ is a random variable following a uniform distribution $U(0.8, 1.2)$.

7. $a_7$ is the zeitgeber time of the second peak phase, $a_7 = \xi_{a7} + a_7^*$. $a_7^*$ is the zeitgeber time of the second peak phase of standard curve, and $\zeta_{a7}$ is a random variable following uniform distribution $U(-1, 2)$.

8. $a_8$ is the minima of the curves, $a_8 = a_4 . f_3(\overrightarrow{a, t}) = a_9 e^{-\frac{(t - a_{11})^2}{2 a_{10}^2}} + a_{12}$.

9. $a_9$ is the amplitude of the third peak, $a_9 = \zeta_{a9}(a_4 + a_1) - a_8$. $\zeta_{a9}$ is a random variable following uniform distribution $U(0.5, 1.5)$.

**Table 1. Convolutional neural network structure.**

|  | neurons | filters | filter size | strides | padding | activation function |
|---|---|---|---|---|---|---|
| convolution layer 1 | $(25-5)\times16$ | 16 | $6\times1$ | 1 | SAME | ReLU |
| convolution layer 2 | $(20-3)\times32$ | 32 | $4\times1\times16$ | 1 | SAME | ReLU |
| convolution layer 2 | $(17-3)\times64$ | 64 | $4\times1\times32$ | 1 | SAME | ReLU |

10. $a_{10}$ is the width of third peak, $a_{10} = \zeta_{a10}a_{10}{}^*$. $a^*{}_{10}$ represents the width of the third peak of the standard curve, and $\zeta_{a10}$ is a random variable following uniform distribution $U(0.8, 1.2)$.

11. $t_{11}$ is the zeitgeber time of the third peak phase, $t_{11} = t_1+\zeta_{a11}$ is a random variable following uniform distribution $U(0, 1)$.

12. $a_{11}$ is the minima of the curves, $a_{11} = a_8$.

Then, we created curves via ODE and Gaussian functions with the above parameters, $\overrightarrow{p}$, and extracted 25 data points at each ZT. We generated 10 W training curves; therefore, training sets can be described as ($Y_{i,j}, i = 1,2,3\ldots25, j = 1,2,3\ldots10^5$). Additionally, a standard normal random perturbation was used to simulate experimental noise. Thus, we created the final training sets, including the circadian patterns (parameter variation) and measurement errors (white noise).

## Convolutional neural network architecture

The CNN in our deep learning model has three convolution layers, two pooling layers and two fully connected layers. Each fully connected layer consists of 1024 neurons. The dropout was inserted into the first fully connected layer to avoid overfitting, and the probability was set (1—drop probability) to 0.5. The parameters of each layer are as specified in Table 1 and Table 2. Loss function, $loss = \frac{1}{n}\sum_{i=1}^{n}(y_{ii} - y_{pi})$

$n$ is the number of training data, $y_{ii}$ and $y_{pi}$ represent the i-th input data and i-th predicted result. Our optimizer was *Adam*, and the learning rate was set to 0.0001.

## Effect size

To quantitate the strength of phenodeviance in mutant mice, we used standardized effect size (Cohen's *d*), $d = \frac{\overline{x_m}-\overline{x_{wt}}}{s}$, to measure the difference between mutant and wild-type mice. $\overline{x_m}$ and $\overline{x_{wt}}$ is mean value of phases or onset times of mutant and wild-type mice in corresponding center. $s$ is the pooled standard deviation, as

$$s = \sqrt{\frac{(n_m - 1)s_m{}^2 + (n_{wt} - 1)s_{wt}{}^2}{n_m + n_{wt} - 2}}$$

where $n_m$ and $n_{wt}$ is the number of mutant and wild-type mice in corresponding center, $s_m$

**Table 2. Convolutional neural network parameters.**

|  | pooling function | pooling size | strides |
|---|---|---|---|
| pooling layer 1 | max_pool | 1x2 | 2 |
| pooling layer 2 | max_pool | 1x2 | 2 |

and $s_{wt}$ is the standard deviation of peak phases and onset times of mutant and wild-type mice in corresponding center. A larger value of effect size always indicates a stronger phenodeviance.

### Metabolic rhythm measure and analysis

Mice were housed in individual metabolic cages in a temperature-controlled animal facility for an adaptation period of 3 days and were continuously recorded for another 3 days in 20 min time bins. The activity and food intake rhythmicities were calculated as described in our previous studies [16,29].

### Locomotor activity analysis

For wheel-running activity assay, as previously described [15], six to ten four-month-old mice were individually housed in cages equipped with running wheels, and they were initially entrained to a LD cycle for at least 7 days, followed by constant darkness for several weeks. To exam light entrainment in mice, a skeleton photoperiod with two light pulses from clock time 07:45 to 08:00 and from clock time 20:00 to 20:15 (ZT0, clock time 08:00) was used. Mice were initially entrained to LD 12:12 for at least 14 d and then released to skeleton photoperiod for 21d [46]. For the jetlag experiments, mice were entrained to a LD 12:12 cycle for 10 days and then LD cycle was advanced 6 hr for 20 days before getting back to the original setting [9]. To assess masking in mice, LD 3.5:3.5 cycle was used. mice were entrained to a LD 12:12 cycle for 10 days, then released to LD 3.5:3.5 cycle for 7 days [47]. Wheel rotation was recorded using ClockLab software (Actimetrics, RRID:SCR_014309).

### RNA isolation, RT-PCR and mRNA expression analyses

RNA isolation and RT-PCR (including primers for mRNA profiling) were carried out as previously described [48]. The relative levels of each RNA were normalized to the corresponding *Actin* levels. Each value used for these calculations was the mean of at least three replicates of the same reaction. Relative RNA levels are expressed as the percentage of the maximal value obtained for each experiment. Each mean ± s.d. was obtained from three biological independent experiments.

### In situ hybridization of the SCN

Mice were euthanized by cervical dislocation at the indicated time points. Coronal sections containing the SCN were processed for in situ hybridization with cRNA sense or antisense probes from nucleotides 581–1412 (NM_011990.1) for *Slc7a11*. Hybridization steps were performed as in our previous study [48].

## Supporting information

**S1 File. Activity/rest scatter plots.**
(RAR)

**S2 File. Food intake scatter plots.**
(RAR)

**S3 File. SyncScreener.**
(RAR)

**S1 Data. The numerical data underlying graphs and summary statistics.**
(XLSX)

**S1 Table. Onset times of wild-type mice from visual assessment.**
(DOCX)

**S2 Table. Peak phases of wild-type mice from visual assessment.**
(DOCX)

**S3 Table. Onset times of wild-type mice from machine learning algorithm.**
(DOCX)

**S4 Table. Peak phases of wild-type mice from machine learning algorithm.**
(DOCX)

**S5 Table. Effect size and p value of known mutants (visual assessment).**
(DOCX)

**S6 Table. Number of mutant lines in each center.**
(DOCX)

**S7 Table. Mutant lines.**
(DOCX)

**S8 Table. Onset times and peak phases of mutant lines.**
(DOCX)

**S9 Table. Mutant lines existing in at least two centers.**
(DOCX)

**S10 Table. Mutant lines for the secondary criteria.**
(DOCX)

**S11 Table. Hits from secondary criteria.**
(DOCX)

**S12 Table. Phenotype associated assay.**
(DOCX)

**S1 Fig. Distribution of the onset time and peak phase in five IMPC centres identified by visual assessment.** Histogram of the onset and peak phase results obtained from five IMPC centres (ICS, WTSI, RBRC, TCP and HMGU) under 12-hour light and 12-hour dark cycles. n = 2201 C57BL/6N mice for activity and n = 2160 C57BL/6N mice for food intake measured by indirect calorimetry over time. Pink column: onset time, red column: peak phase.
(TIF)

**S2 Fig. Onset time and peak phase identification in synthetic training sets.** (A) Mean data of food intake for wild-type mice from HMGU as an example. Red dots are averaged raw data and multicoloured curve is the fitted curve. Part data before peak phase (data between two red dashed lines) are fitted by a piecewise function in (B).Blue curve is the fitting curve. Onset is identified as the first data entering transition (green arrow), while peak phase is at the end of transition (blue arrow). Other data in (A) are fitted by Gaussian functions (green curves). (B) Piecewise function for fitting data points before E peak phase. Dividing points of two stages are indicated by two black arrows in (A) and (B).
(TIF)

**S3 Fig. Bland–Altman plots of the bias between the two methods, the visual and SyncScreener for activity.** The 95% limits of agreement (1.96 s.d.) were calculated to determine whether

the SyncScreener could replace visual assessment. Activity onset data and peak phase activity data obtained by the five centres (ICS, WTSI, RBRC, TCP and HMGU).
(TIF)

**S4 Fig. Bland–Altman plots of the bias between the two methods, the visual and SyncScreener of food intake.** Food intake onset data and peak phase data obtained by the five centres (ICS, WTSI, RBRC, TCP and HMGU).
(TIF)

**S5 Fig. Daytime activity and food intake of *Oxtr*^*tm1.1/tm1.1*^ mice.** (A and B) Ratios of daytime activity (A) and food intake (B) to that at night were calculated using data from *Oxtr*^*tm1.1/tm1.1*^ and wild-type mice data from RBRC. Two-way ANOVA was employed to test the statistical significance. ****: $P < 0.0001$.
(TIF)

**S6 Fig. Generation of *Slc7a11*^*tm1a/tm1a*^ mice.** (A) Schematic of knockout strategy for *Slc7a11* based on knockout-first design. (B) Forward and reverse primers for genotyping. (C) PCR analysis of tail genomic DNA for wild-type and *Slc7a11*^*tm1a/tm1a*^ alleles in wild-type, heterozygous and homozygous knockout mice.
(TIF)

**S7 Fig. Profiles of energy expenditure parameters using a CLAMS.** (A) food intake; (B) the volume of CO2; (C) the volume of O2; (D) the heat. Rhythms of food intake, VCO2, VO2 and the heat were plotted over a 72 hr time frame as the mean ± SEM (n = 10). Two-way ANOVA was used to determine the statistical significance, *$p < 0.05$.
(TIF)

**S8 Fig. Reentrainment of *Slc7a11*^*tm1a/tm1a*^ mice to a new light-dark cycle.** (A) Representative actograms of wheel-running activity of wild-type and *Slc7a11*^*tm1a/tm1a*^ mice subjected to a 6-hr phase advance and delay in LD cycle. At day 22, the recording was disrupted for about 24 hours. (B and C) Re-entrainment traces of phase advance (B) and delay (C) of wild-type (Blue) and *Slc7a11*^*tm1a/tm1a*^ (Red) mice. n = 9 for wild-type mice, n = 5 for *Slc7a11*^*tm1a/tm1a*^ mice. Two-way ANOVA was employed to test the statistical significance. *: $P < 0.05$.
(TIF)

**S9 Fig. Masking of wild-type and *Slc7a11*^*tm1a/tm1a*^ mice during LD 3.5:3.5.** (A and B) Representative actograms of daily wheel-running activity of wild-type (A) and *Slc7a11*^*tm1a/tm1a*^ mice (B). Light phases are indicated in yellow to show the structure of the LD 3.5:3.5 cycle as well as to help visualize the occurrence of wheel-running activity under this schedule. (C) Masking ratios of wild-type and *Slc7a11*^*tm1a/tm1a*^ mice during LD 3.5:3.5, which are calculated by dividing total activity during light phases with that during dark phases. n = 7 for each genotype. Two-way ANOVA was employed to test the statistical significance. n.s.: $P > 0.05$.
(TIF)

**S10 Fig. Expression profiles of the core clock genes in the liver tissues.** Error bars represent the s.d. for each time point from three biological independent replicates. Two-way ANOVA was employed to test the statistical significance. n.s.: $P > 0.05$.
(TIF)

## Acknowledgments

The authors thank all IMPC members and partners for their contribution to the consortium effort and thank members of Cambridge -Suda GRC for their assistance in animal facility and members

of the Xu laboratory for discussion. Sanger Institute Mouse Genetics Project Members are as follows: David Lafont, Valerie E. Vancollie, Robbie S.B. McLaren, Emma Sanderson, Christine Rowley, Mark Griffiths, Brendan Doe, Nicola Cockle, Joanna Bottomley, Edward Ryder, Diane Gleeson, Ramiro Ramirez-Solis, Hannah Wardle-Jones, David J. Adams, Graham Duddy

## Author Contributions

**Conceptualization:** Tao Zhang, Ying Xu.

**Data curation:** Tao Zhang, Pancheng Xie, Zhengyun Huang, Qiaocheng Zhai, Ling Yang, Ying Xu.

**Formal analysis:** Pancheng Xie, Yue Gu, Ling Yang, Ying Xu.

**Funding acquisition:** Yingying Dong, Zhiwei Liu, Ling Yang, Ying Xu.

**Investigation:** Tao Zhang, Pancheng Xie, Zhengyun Huang, Qiaocheng Zhai, Yi Liu, Ling Yang, Ying Xu.

**Methodology:** Tao Zhang, Ling Yang, Ying Xu.

**Project administration:** Ying Xu.

**Resources:** Tao Zhang, Pancheng Xie, Yingying Dong, Zhiwei Liu, Fei Zhou, Dejing Pan, Nobuhiko Tanaka, Yuichi Obata, Allan Bradley, Christopher J. Lelliott, Lauryl M. J. Nutter, Colin McKerlie, Ann M. Flenniken, Marie-France Champy, Tania Sorg, Yann Herault, Martin Hrabe De Angelis, Valerie Gailus Durner, Ann-Marie Mallon, Helen E. Parkinson, Jun Yan, Jan Rozman.

**Software:** Tao Zhang, Ling Yang.

**Supervision:** Ying Xu.

**Validation:** Tao Zhang, Pancheng Xie, Zhengyun Huang, Qiaocheng Zhai, Ling Yang, Ying Xu.

**Visualization:** Tao Zhang, Pancheng Xie, Ying Xu.

**Writing – original draft:** Tao Zhang, Yi Liu, Ying Xu.

**Writing – review & editing:** Tao Zhang, Qingyu Wu, Ann M. Flenniken, Yann Herault, Ann-Marie Mallon, Steve D. M. Brown, Terry Meehan, Helen E. Parkinson, Damian Smedley, K. C. Kent Lloyd, Xiang Gao, Je Kyung Seong, Chi-Kuang Leo Wang, Radislav Sedlacek, Yi Liu, Jan Rozman, Ying Xu.

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
