## [Decision Letter · Decision Letter 0]

10 Sep 2019

Dear Dr Xu,

Thank you very much for submitting your Research Article entitled 'High-throughput discovery of genetic determinants of circadian entrainment' to PLOS Genetics. Your manuscript was fully evaluated at the editorial level and by independent peer reviewers. The reviewers appreciated the attention to an important problem, but raised some substantial concerns about the current manuscript. Based on the reviews, we will not be able to accept this version of the manuscript, but we would be willing to review again a much-revised version. We cannot, of course, promise publication at that time.

If you decide to revise the manuscript for further consideration at PLOS Genetics, please aim to resubmit within the next 60 days, unless it will take extra time to address the concerns of the reviewers, in which case we would appreciate an expected resubmission date by email to plosgenetics@plos.org.

[LINK]

We are sorry that we cannot be more positive about your manuscript at this stage. Please do not hesitate to contact us if you have any concerns or questions.

Yours sincerely,

Achim Kramer

Associate Editor

PLOS Genetics

Gregory Barsh

Editor-in-Chief

PLOS Genetics

Reviewer's Responses to Questions

**Comments to the Authors:**

Reviewer #1: Zhang et al. developed a high-throughput screening approach to identify genes involved in circadian entrainment in mice. They use WT activity and feeding data from five different IMPC centers as reference data set and test this against data from 750 mutant lines. They identify 5 new mutants of which one is followed up. Slc7a11 mutant mice show advanced activity onsets and some changes in the expression of cell coupling genes in the SCN.

The paper describes an interesting and very promising approach - and a clever use of existing consortium data. I have two major points of criticism:

1. Maybe I misunderstood this, but to me it is somewhat surprising that from the 750 screened lines no known clock regulators with activity phenotypes have emerged.

2. While the proclaimed target of the screen is circadian entrainment only LD12:12 activity and gene expression data are shown, and no entrainment-targeted experiments were performed.

Further points:

1. It is surprising that the activity phase advance for the Slc7a11 mutants is rather mild and not very robust. Was this one of the best mutants? How often were strong onset advances or delays (+/- 3 hrs or more) observed?

2. Line 103 – would suggest inserting “but may also promote” before “sleep disorders” as chronodisruption is not the only cause for the following disorders.

3. Line 147 – make sure to clearly distinguish between “circadian” and “diurnal”. Since you are studying LD conditions in most cases, “daily” or “diurnal” would be correct here.

4. Page 9 – why were activity data binned at 1-h intervals? For the machine learning algorithm this should not make much of a difference and you are simplifying the data, thus potentially masking interesting phenotypes.

5. Line 193 – you argue that you excluded M peaks from analysis because they are less robust, but what would that mean for entrainment? Entrainment phase should be reflected in both peaks. I guess there is a bit of a misconception between “entrainment” and “activity pattern/profile” throughout the paper. Along this line it does not surprise that SCN clock gene expression is not consistently altered in the Slc7a11 mutant since effects could be downstream of the SCN.

6. Line 265 – what do you mean by “similar phenotypes”?

7. Page 14 – to assess entrainment I would expect T-cycle experiments or experiments under different light intensity conditions. Could the advances of activity onsets in the Slc7a11 mutants be explained by altered light masking? Do the mice entrain faster/slower under shifted LD cycle conditions?

8. Figs. 3 & 5 – relabel “ZT30” to “ZT6”

9. Fig. 7C – why was gene expression not tested in the SCN? How were CTs determined for arrhythmic Per1/2 and Bmal1 mutants? It is very counterintuitive that Slc7a11 gains rhythmicity in the two clock-less mutants compared to WT animals.

10. Fig. 7D – clock gene expression appears very much dampened already in the WT animals. Why were genes with very little rhythmicity in the SCN (Clock, Cry2) tested, but not high-amplitude genes such as Dbp or Nr1d1?

11. Fig. 7E – this figure is confusing to me. Several of the tested genes are not rhythmic in the WT SCN. Does this mean the effect of Slc7a11 is independent of the clock? If so, can one then speak of “circadian entrainment” (see also above)?

Reviewer #2: Zhang et al. analyze indirect calorimetry data form IMPC centers to extract circadian entrainment data. Overall this is paper carries out good analysis of existing data to extract new phenotypes. I certainly think it deserves publication in PLOS Genetics. I have several recommendations that will improve clarity of the paper.

Major – sex is not addressed in the paper. This is a major factor that affects many diurinal/circadian parameters, particularly during estrus (https://www.ncbi.nlm.nih.gov/pmc/articles/PMC4288375/). The authors should model sex in their analysis.

The method in which mutants were identified is not clear or explained (pvalue, effect size, and variance are combined to find 5 affected mutants. The process is arbitraty.

The machine learning based method developed here (SyncScreener) it is not stated how and where this will be distributed.

CAM-SU Genomic Resource Center is cited as receiving S1 File and S2 File. There is no information on how the reader can access this. If this is not a maintained data repository please deposit the data at other well curated databases.

Will the analysis performed here be part of the mousephenotype.org (impc) website? Given these are IMPC strains, it would be good to have this data available on this site for the community.

Specific comments –

Figure 1 – I didn’t find this figure useful. It outlines the flow of the paper but does not add any information. It also uses terms that are confusing and misleading.

For instance – “Discovery of genetic determinants” is genotype association. The terms primary and secondary screen makes it seem as if there were two levels of screening. This should state something like primary criteria and secondary criteria. “Validation” was done by regenerating mutant lines for wheel running or longer IC studies. This is not conveyed in the figure. Similarly, the figure legend lacks description.

In general, a figure outlining the process of IC at the various centers will be more useful. Emphasize the differences between the centers. Even though the protocols are available online, the paper should concisely state the detailed protocol at the centers and emphasize the differences (light cycle etc.) between the centers. If not in this figure, it should be a supplementary figure. This will help orient the reader on the challenges in analyzing this data.

Minor – typo in figure. “Cellecting” should probably be “collecting”.

Figure 2 –

A, B - The data is only shown for 18 hours (x-axis). The text does not clearly explain why. I’m assuming that this the general length of the IC protocol. Most reader will expect a 24 hr LD bar. HMGU is padded on the right with 0, please explain this.

The data is sorted on the y axis (mice seem to be organized by peak phase), however, nothing is stated about this in the figure legend.

Page 9, line 180 states “moreover, these two parameters exhibited a stable phase relationship…” I’m not sure what is meant by a stable phase relationship. Do you mean the phase relationship is consistent across centers or within a center or both. They certainly vary across centers – the difference between Activity peak phase and onset varies from 1 hr to 3hr. Please clarify these for the reader.

Line 197-199 describe LD conditions and are not in the methods. See earlier point about describing experimental differences between centers.

Figure 3 – why are the food intake pattens not shown for Fbxl3 and Zbtb20?

Which center was the validation data generated?

Figure 4 – this is a key figure that describies finding the mutants using the methods developed above. I find that a clear figure showing the distribution of the 5 final mutants is missing. I would like to see the phenotype(s) of these plotted against controls or other mutants. The “secondary screen” is actually just a secondary criteria set that the original mutants are placed through. Please explain the rationale for these – why are all lines that have a greater than 2sd effect size selected? Why not use pvalue from the onset regardless of the effect size. The rationale for comparison is not clear. Effect size is used in primary and then a combination of effect size, pvalue (strict at 0.001), and variance (50% must have significant phenotype) is used in the secondary critera. This seems like conditions were tweaked till an acceptable number of hits arose. This should be clarified.

Just to make sure that the baby is not thrown out with the bathwater - do the 88 genes that survive primary analysis show enrichment in certain pathways or gene ontologies. Do they show significant enrichment of human GWAS hits for chronotype phenotypes? There has been a slew of these papers.

Visualization point - The 3D barplots are very hard to interpret. I found myself drawing lines to in 3D space to compare the right bars (https://guides.library.duke.edu/datavis/topten). Showing the data in multiple 2D histograms will be more interpretable. Small multiple plots are generally easier to interpret (https://www.r-bloggers.com/why-you-should-master-the-small-multiple-chart/ or https://journals.plos.org/ploscompbiol/article?id=10.1371/journal.pcbi.1003833 (Rule 8)).

Figure 5 – describes the phenotypes of the 5 selected mutants. Please provide a table similar to Figure 3G table for these mutants that clearly describes the effect size and pvalue (table S11 does not provide both stats).

How do these mutants compare in effect size and pvalue to the validation mutants for the same phenotypes. The Slc7A11 phenotype seems to have a very low effect. It would be helpful to have E and M peaks clearly marked for control and mutant on these plots.

Figure 6 – The validation of Slc7a11 confirms that this is a very subtle mutant. The onset of activity is slightly disorganized, perhaps advanced. The wheel running DD data has no effect, when mutations with phase advance usually have shorter period. The wheel running actogram that is shown shows that there is slightly more daytime activity in the mutants. This lack of activity consolidation should lead to lowered circadian amplitude. Please check this.

Figure 7 – no comments.

Reviewer #3: This work takes data collected by the IMPC and analyses it for the timing of behavior. This is a wonderful idea, taking advantage of a resource that has many treasures still to be discovered and extending the usefulness of the massive amount of work that has gone into building this program. The validation of the candidate gene, Slc7a11, is also very nicely handled. I do have several comments which I think will make the article more useful to a general readership and boost the scholarship as well.

“However, how internal circadian clocks are entrained to changes in photoperiod remains unclear. “

This sentence implies that changes in photoperiod will be addressed. They are not. The authors rather use entrained phase as a phenotypic marker in their screen. Another level of complexity would be to screen on entrainment to different photoperiod.

“In addition, activity, feeding, temperature and glucocorticoid signals can also affect the circadian phase of the circadian clock [17-21]. These studies indicate that circadian entrainment is influenced at multiple regulatory levels.”

I think it would be better to say that these are all zeitgebers of the circadian clock and thus by definition, they will impart phase information on their target tissues. Not all of the papers cited actually discuss or systematically probe phase (though they do show phase shifting at least).

“We hypothesized that mouse mutants with impaired circadian entrainment would result in phase changes in locomotor activity and/or food intake behaviour under the light/dark cycle “

Why? The authors should put this hypothesis into a better context. The story of food entrainment – the one before the molecular era – is inspiring to all scientists! It is an opportunity to help non-chronobiologists understand that chronobiology is not just a time point in the dark and a time point in the light, that it is a robust machinery that regulates our behaviour and physiology systematically. Further, PLoS Genetics is a non-clocks journal and the concept (peripheral oscillators etc.) would benefit from a graphical explanation as well as a few words here.

“The onset time was defined as a transition from rest to steep activity/food intake, while the peak phase was the end of the transition.”

This is not clear. Which point on the transition? For dim light melatonin onset, many groups empirically decide which point of the onset (e.g. 25% level of the upslope of onset) to use.

„The results for the onset time and peak phase of activity and food intake for each centre were determined by SyncScreener (Figs 2C and 2D, S3 and S4 Tabl “

I think it would be worth graphing the peak and onset values for activity and food intake for the 5 centers. The data are shown here and we get an impression of the differences but we should see this graphed rather than these figures and the table in the suppl materials. Graphical representation would be helpful to appreciate the variance.

“We reasoned that the onset time was easily recognized by visual assessment due to an obvious steep ascension from an inactive state, whereas peak phase sometimes displayed as a plateau, which may lead to variability in peak phase identification.”

Therefore the onsets are used? It is not clear what the conclusion is here and if a decision was taken. If the peak is unstable (I have seen this also and also prefer onsets) a measure called Center of Gravity can also be used.

“Finally, to validate our predictive models and parameters, we used several known circadian mutant lines to evaluate the circadian parameters”

Two things here: Before going into this, we would like to know about the variance (beyond the nice heat maps) in the controls. How does this look in comparison to wheel running data in the same strain? I think it would be important to not just show the pooled data from the sites but to show the variation of all individuals. What is the onset with SD? Before looking at the mutants, we would like to know this.

How does this data on clock mutants compare to what is published?

“delayed onset and peak phase of both food intake and activity compared to wild-type mice at TCP (Figs 5C-5F), suggesting that circadian robustness might be impaired.“

I do not agree with this statement/conclusion. Can the authors back it up?

„Rhbdl1+/tm1a mutant mice also displayed vision defects“

Very nice observation showing the power of this screen to reveal entrainment mutants. (In Neurospora there have been mutants reported with no difference in period but rather in entrainment.)

“The Oxtrtm1a/tm1a mice exhibited a trend towards more daytime activity and food intake tendency”

Is this backed up somewhere with data?

**Have all data underlying the figures and results presented in the manuscript been provided?**

Reviewer #1: Yes

Reviewer #2: No: the supplementary data 1 and 2 need to be deposited to a curated database. the code and neural network weights should be shared on github or similar repository.

Reviewer #3: Yes

PLOS authors have the option to publish the peer review history of their article (what does this mean?). If published, this will include your full peer review and any attached files.

Reviewer #1: No

Reviewer #2: No

Reviewer #3: No

---

## [Decision Letter · Decision Letter 1]

19 Dec 2019

Dear Dr Xu,

We are pleased to inform you that your manuscript entitled "High-throughput discovery of genetic determinants of circadian misalignment" can be principally accepted for publication in PLOS Genetics pending that you deal with the one remaining minor concern of Reviewer #3.

In addition, before your submission can be formally accepted and sent to production you will need to complete our formatting changes, which you will receive in a follow up email. Please be aware that it may take several days for you to receive this email; during this time no action is required by you. Please note: the accept date on your published article will reflect the date of this provisional accept, but your manuscript will not be scheduled for publication until the required changes have been made.

Yours sincerely,

Achim Kramer

Associate Editor

PLOS Genetics

Gregory Barsh

Editor-in-Chief

PLOS Genetics

Comments from the reviewers (if applicable):

Reviewer's Responses to Questions

**Comments to the Authors:**

Reviewer #1: I thank the authors for addressing my suggestions and revising the paper. I has much improved.

Reviewer #3: Thank you for the extensive revisions in response to the reviewer feedback. I have one single remaining objection, namely to the sentence " However, how internal circadian clocks are entrained to changes in photoperiod remains unclear." The paper does not address this issue. The addition of a skeleton photoperiod is a method to probe entrainment but it does not address alternative photoperiods. My impression is that entrainment was used to classify mutant phenotypes and I would suggest using this more encompassing term instead.

**Have all data underlying the figures and results presented in the manuscript been provided?**

Reviewer #1: Yes

Reviewer #3: Yes

PLOS authors have the option to publish the peer review history of their article (what does this mean?). If published, this will include your full peer review and any attached files.

Reviewer #1: No

Reviewer #3: No

**Data Deposition**

http://datadryad.org/submit?journalID=pgenetics&manu=PGENETICS-D-19-01158R1

**Press Queries**

---

## [Editor Report · Acceptance letter]

3 Jan 2020

PGENETICS-D-19-01158R1 

High-throughput discovery of genetic determinants of circadian misalignment 

Dear Dr Xu, 

We are pleased to inform you that your manuscript entitled "High-throughput discovery of genetic determinants of circadian misalignment" has been formally accepted for publication in PLOS Genetics! Your manuscript is now with our production department and you will be notified of the publication date in due course.

With kind regards,

Matt Lyles

PLOS Genetics

On behalf of:
